# ROYAL SOCIETY
# OPEN SCIENCE

analysis/biomedical engineering

moment of velocity, instantaneous frequency, Hilbert transform, electrocardiogram, bio-signal processing

**Author for correspondence:**
M. Dorraki
e-mail: mohsen.dorraki@adelaide.edu.au

# On moment of velocity for signal analysis

M. Dorraki[1,2], A. Fouladzadeh[3,4], A. Allison[1,2], B. R. Davis[1] and D. Abbott[1,2]

[1]School of Electrical & Electronic Engineering, and [2]Centre for Biomedical Electrical Engineering (CBME), The University of Adelaide, Adelaide, South Australia 5005, Australia
[3]Centre for Cancer Biology, University of South Australia and SA Pathology, Adelaide, South Australia, Australia
[4]Department of Surgery & Tumour Immunotherapy Laboratory, University of Adelaide, Royal Adelaide Hospital, Adelaide, South Australia 5005, Australia

MD, 0000-0002-6675-3393; AF, 0000-0002-3981-4509;
AA, 0000-0003-3865-511X; BRD, 0000-0002-5542-5909;
DA, 0000-0002-0945-2674

The instantaneous frequency (IF) of a signal is a well-defined quantity that is widely used for analysing non-stationary signals. However, often in practice, IF as a function of time can possess large spikes and negative values. Moreover, IF is very sensitive to noise, limiting its range of practical application. Due to these deficiencies, we introduce the concept of moment of velocity (MoV) for signal analysis. As a case study, we compare the performance of MoV to a standard Hilbert transform-based approach for R-wave identification in human electrocardiogram signals, demonstrating that our approach is more robust to noise. We examine characteristic heartbeats obtained from the MIT-BIH Arrhythmia database. A detection error rate of 0.07%, a positive predictive value of 99.97%, and a sensitivity of 99.95% are achieved against analysis results from the database.

## 1. Introduction

The concept of the Hilbert transform and instantaneous frequency (IF), amplitude and phase is used in many scientific applications, such as communications, seismology, sonar and biomedical engineering [1,2]. Conceptually, IF is explicated as the frequency of a sine function that locally fits the signal. The definition of IF is based on the Hilbert transform and the phase derivative of the analytic signal [3].

The application of IF is limited for multicomponent signals. Note that the IF of a signal normally is a fluctuating waveform, which is sensitive to the amplitudes of the frequency components. Therefore, the interpretation of spectral information is often severely constrained by the nonlinear nature of the IF.

In order to overcome this difficulty, a number of useful quantities are introduced in the literature. To control the impact

of the amplitudes, the concept of weighted average IF (WAIF) can be used for multicomponent signals [4]. The intensity and envelope weighted average of IF (IWAIF and EWAIF) are introduced to increase computational efficiency and accuracy [5]. The main difference between the IWAIF of a signal and its EWAIF is the choice of weighting function. Both IWAIF and EWAIF are constant and independent of time. Therefore, they are not appropriate for a signal with varying frequency such as a chirp. One useful concept for describing the changing spectral structure of a time-varying signal is its average frequency as a function of time, which arises in time-frequency distribution theory and is the first conditional moment of frequency or conditional mean frequency of the distribution [6,7].

In this paper, we suggest the concept of moment of velocity (MoV) as a convenient tool for analysis of non-stationary signals. As will be seen, MoV is very similar to IF except that it is more robust to noisy conditions. We show that MoV can suppress the large spikes that often clutter the IF signal. Therefore, it can provide spectral information of the original signal in a more convenient way. As a case study application of MoV, we demonstrate its use in electrocardiogram (ECG) signals from the MIT-BIH Arrhythmia database [8] in order to identify the QRS complex in ECG waveforms.

# 2. The Hilbert transform

Many approaches have been presented for signal instantaneous parameters estimation such as those based on Gabor's method, EMD, the Wigner distribution, etc. In this section, we will briefly review the Gabor approach that uses the Hilbert transform to produce the IF.

## 2.1. The Hilbert transform equations

Any one-dimensional integral transform pair may be written as $u(t) \Leftrightarrow U(s)$ [3], where a time (or other variable) function $u(t)$ is transformed into a complex function of a real or complex variable $s$. The transform pair is defined by the following pair of integrals:

$$U(s) = \int_{\Omega} u(t)\varphi(t, s)\,\mathrm{d}t; \quad t \in \Omega \tag{2.1}$$

and

$$u(t) = \int_{\Gamma} U(s)\Psi(s, t)\,\mathrm{d}s; \quad s \in \Gamma, \tag{2.2}$$

where the function $\varphi(t, s)$ is the kernel; $\Psi(s, t)$ is the conjugate kernel; $U(s)$ is called the (integral) transform of $u(t)$; and $u(t)$ is the inverse transform. The Hilbert transform pair is defined by the integrals [9]

$$U(s) = \frac{1}{\pi}\int_{-\infty}^{\infty} \frac{u(t)}{s - t}\,\mathrm{d}t; \quad -\infty < t < \infty \tag{2.3}$$

and

$$u(t) = \frac{1}{\pi}\int_{-\infty}^{\infty} \frac{U(s)}{t - s}\,\mathrm{d}s; \quad -\infty < t < \infty. \tag{2.4}$$

A comparison with the general forms of (2.1) and (2.2) shows that the Hilbert transform is defined using the kernel $\varphi(t, s) = 1/[\pi(s - t)]$ and the conjugate kernel $\Psi(s, t) = 1/[\pi(t - s)]$; that is, the kernels differ only by sign. The variable $s$ is a time variable. Therefore, the Hilbert transform of a function of time is another function of a time of different shape. The mathematical definition of the Hilbert transform is usually written in the form

$$y(t) = \mathrm{H}[x(t)] = \frac{1}{\pi}\int_{-\infty}^{+\infty} \frac{x(\tau)}{t - \tau}\,\mathrm{d}\tau. \tag{2.5}$$

The Hilbert transform can be considered as a convolution between $1/\pi t$ and the signal $x(t)$, i.e. $y(t) = x(t) \otimes 1/\pi t$, and the inverse relation is $x(t) = y(t) \otimes -1/\pi t$.

## 2.2. Instantaneous parameters

The complex signal that has an imaginary term equal to the Hilbert transform of the real part is called the analytic signal $z(t)$, where $z(t) = x(t) + j\mathrm{H}[x(t)]$. The instantaneous amplitude is defined as the magnitude of the analytic function or the instantaneous amplitude is absolute value of analytic signal $e(t) = \sqrt{x(t)^2 + \mathrm{H}[x(t)]^2}$. The IF of a signal is a function of time and also a measure of the frequency

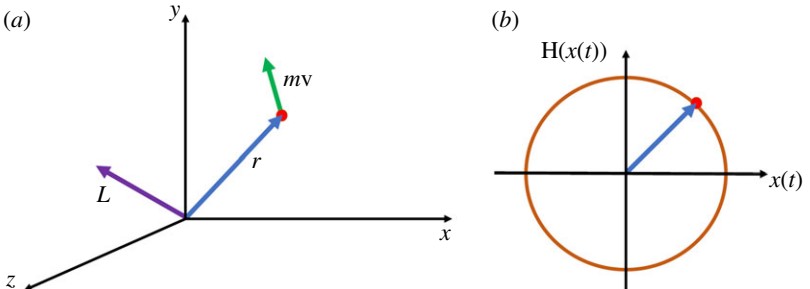

**Figure 1.** (a) Angular momentum coordinate system and (b) signal and the Hilbert transform analytic plane for $x(t) = \sin(t)$.

corresponding to a particular time component of the signal. For a real signal, $s(t)$, the instantaneous frequency, $f(t)$, is defined as

$$f(t) = \frac{1}{2\pi}\frac{\mathrm{d}\phi(t)}{\mathrm{d}t}, \tag{2.6}$$

where $\phi(t)$ is the instantaneous phase of analytic signal, and defined by $\phi(t) = \arctan(\mathrm{H}[x(t)]/x(t))$. Therefore, the IF may be written in following form:

$$
\begin{aligned}
f(t) &= \frac{1}{2\pi}\frac{\mathrm{d}}{\mathrm{d}t}\left[\arctan\left(\frac{\mathrm{H}[x(t)]}{x(t)}\right)\right]\\
&= \frac{x(t)(\mathrm{dH}[x(t)]/\mathrm{d}t) - \mathrm{H}[x(t)](\mathrm{d}x(t)/\mathrm{d}t)}{x(t)^2 + \mathrm{H}[x(t)]^2}.
\end{aligned}
\tag{2.7}
$$

In order for IF to be meaningful (i.e. always non-negative), the slope of the instantaneous phase must be always positive. This implies that the time signal must be locally symmetric to the zero mean. Therefore, as explained in appendix A, any DC offset produces results with negative frequencies that are difficult to interpret.

# 3. The moment of velocity

The initial concept of MoV is derived by analogy from angular momentum in the field of particle dynamics. For an analytic signal, the imaginary part is the Hilbert transform of its real part that is the original signal. Therefore, the original signal is always orthogonal to the Hilbert transform of the waveform. Similarly, the angular momentum vector is perpendicular to the position vector. The angular momentum coordinate system is shown in figure 1a. As an example, a signal and the Hilbert transform analytic plane are illustrated in figure 1b for the case when $x(t) = \sin(t)$.

The angular momentum for a moving particle of unit mass is defined as

$$\mathbf{L} = \mathbf{r} \times \mathbf{v}, \tag{3.1}$$

where $\mathbf{r}$ is the position vector of particle and $\mathbf{v}$ is particle velocity. If the vectors $\mathbf{r}$ and $\mathbf{v}$ are resolved into components and the cross-product formula is applied we obtain,

$$\mathbf{L} = \begin{vmatrix} \mathbf{i} & \mathbf{j} & \mathbf{k} \\ r_x & r_y & r_z \\ v_x & v_y & v_z \end{vmatrix} = (r_y v_z - r_z v_y)\mathbf{i} + (r_z v_x - r_x v_z)\mathbf{j} + (r_x v_y - r_y v_x)\mathbf{k}, \tag{3.2}$$

where $\mathbf{i}$, $\mathbf{j}$, $\mathbf{k}$, are the standard unit vectors, and the indices represent the $x$, $y$ and $z$ components of the vectors. Assuming the particle moves in the $x$-$y$ plane, the angular momentum is perpendicular to the $x$-$y$ plane and equation (3.2) is reduced to only its $z$ components defined by the scalar

$$
\begin{aligned}
L_z &= r_x v_y - r_y v_x\\
&= r_x \frac{\mathrm{d}r_y}{\mathrm{d}t} - r_y \frac{\mathrm{d}r_x}{\mathrm{d}t}.
\end{aligned}
\tag{3.3}
$$

In terms of signal and the Hilbert transform of the analytic signal, we replace $r_x$, $r_y$ and $L_z$ in equation (3.3) by $x(t)$, $\mathrm{H}(t)$ and MoV, respectively. Therefore, MoV is defined as

$$\text{moment of velocity } = x(t)\frac{\mathrm{dH}[x(t)]}{\mathrm{d}t} - \mathrm{H}[x(t)]\frac{\mathrm{d}x(t)}{\mathrm{d}t}. \tag{3.4}$$

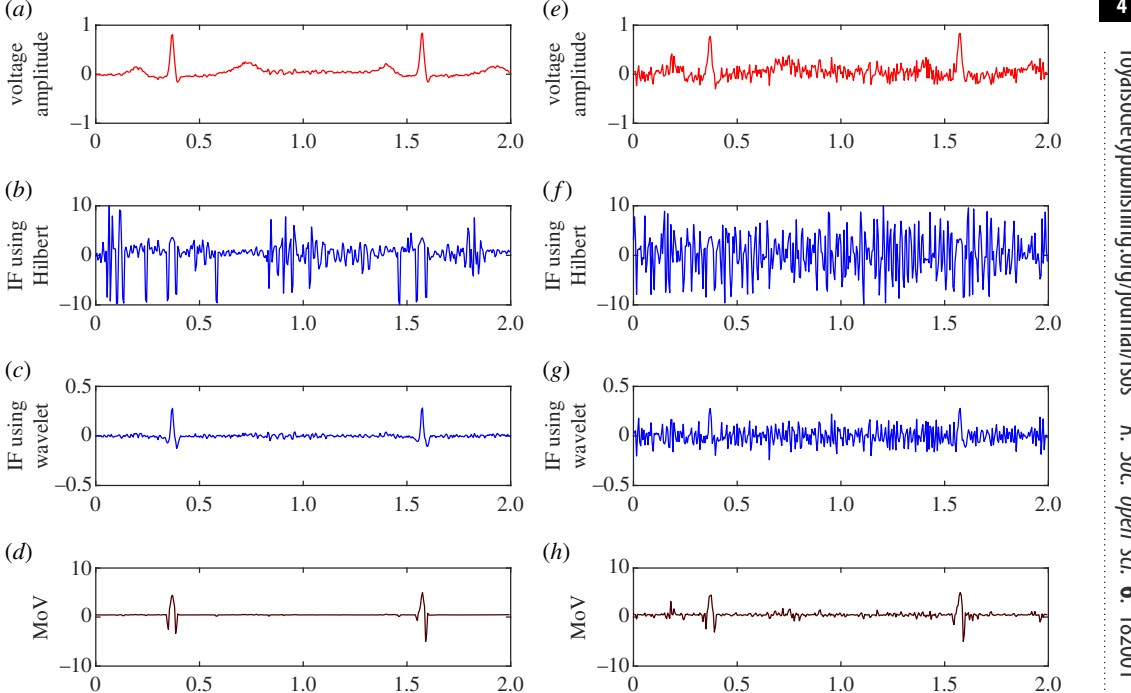

**Figure 2.** For qualitative comparison of IF and MoV, (*a*) the characteristic heartbeat, (*b*) the corresponding instantaneous frequency obtained from the Hilbert transform, (*c*) instantaneous frequency obtained from complex wavelet and (*d*) MoV, are shown for a normal patient. On the right (*e–h*), we add 20 dB SNR Gaussian white noise to the ECG in order to illustrate the effect of noise.

# 4. Case study: moment of velocity in ECG analysis

In this section, as a case study, we illustrate the application of MoV in the analysis of an ECG time series. To demonstrate the utility of MoV, an ECG time series is selected as it is a well-known class of non-stationary signal.

The ECG waveform, which reveals the electrical activity of the heart, is a vital physiological signal [10]. Note that ECG signals are transient, time-varying and non-stationary. The fact that ECG signals are non-stationary imposes a number of limitations on using basic time and frequency analyses. Most time and frequency analyses are established on the assumption that signals are stationary or locally stationary. Therefore, these methods cannot cope with all the characteristics of ECG signals. Consequently, Hilbert transforms have been found to be useful for analysis of an ECG signal (e.g. [11–15]). The combination of some useful properties of the Hilbert transform and other mathematical tools is used for detecting the QRS complex in the study of Benitez *et al.* [16]. Current methods have varying success in their ability to determine R peaks. Not only do ECG signals vary in morphology, baseline, amplitude and signal strength but also they are usually contaminated with noise, and the analysis of an ECG signal with noise is challenging. We employ the detection of R peaks as a case study to demonstrate the utility of MoV.

## 4.1. R-wave detection using the moment of velocity

In an ECG waveform, the interval between cardiac cycles is called the beat-to-beat interval and is used to explore instantaneous heart rate [17]. Basically, the beat-to-beat interval is labelled using the letters P, Q, R, S and T for the individual peaks and this is carried out for whole recording [18]. According to the literature [19,20], the most important information about an ECG signal is contained in the P wave, QRS complex and T wave. A Hilbert transform-based approach using IF has been proposed to analyse ECG signals [21]. The fact that the R wave lies in a different frequency range is used as a key factor for QRS identification.

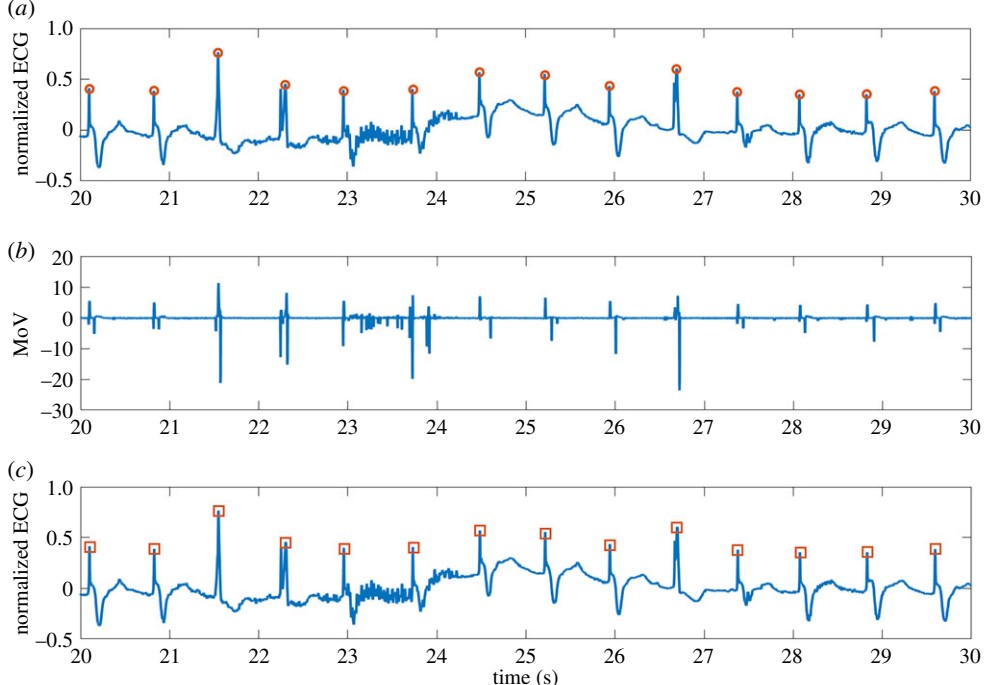

**Figure 3.** (*a*) Characteristic heartbeat of subject 104 from the MIT-BIH Arrhythmia database [8], (*b*) corresponding MoV and (*c*) ECG with the R wave identified using our approach based on MoV.

Figure 2 illustrates the ECG, corresponding IF and MoV curves for a patient from the Physionet database [8]. Here, IF is obtained from two different approaches, the Hilbert transform and the complex wavelet. The particular complex wavelet method we use is based on the Morse wavelet [22,23]. Additionally, 20 dB SNR noise is intentionally added to the signal and the corresponding IF and MoV waveforms are shown. In order to directly compare IF and MoV, no filtering or pre-processing steps are carried out on the ECG.

Considering figure 2, although the IF waveforms possess the frequency information of the ECG signal under analysis, the information is cluttered with large spikes and negative values, particularly in the case of the IF obtained from the Hilbert transform. The influence of DC offset, riding waves and abrupt changes of signal are the main causes of these large spikes and negative IF values. The use of MoV can overcome this problem by removing the numerator that can take on very small values.

As is mentioned in [21], frequency characteristics may be used to identify the QRS complex and the R wave may be identified by detecting spikes in instantaneous frequency plot. However, this is problematic as the IF signal is cluttered with too many spikes as can be seen in figure 2b. Considering figure 2b,c,d, when focusing on higher frequency components or rapidly changing features, the MoV may be a more practical tool than IF.

The R peaks are distinguishable in the MoV plots. In addition, ECG signals may be contaminated by different kinds of noise [20], and all ECG analysis applications require the accurate detection of the R wave in the presence of noise. In figure 2e, we intentionally add 20 dB SNR Gaussian white noise to the ECG signal to show the deleterious impact on IF. In figure 2f,g, it may be seen from corresponding IF plots that the effect of noise obscures any signal features including R waves. Moreover, figure 2h shows the MoV for the same EGC waveforms with 20 dB SNR of added white Gaussian noise, showing that the R waves are maintained above the noise.

After applying MoV to the ECG signal, a simple filtering and peak detection algorithm is used to identify the R waves. The peak detection algorithm must be adaptive, because if the amplitude of the signal varies, the algorithm will miss the R waves that are less than the defined tolerance.

Figures 3 and 4 demonstrate two normalized heartbeat characteristics of 10 s from patients 104 and 122 with manually annotated R waves. The corresponding MoV curves are shown in figures 3b and 4b. The R waves detected using the algorithm are demonstrated in the lower panels, and they correctly match the manual annotations.

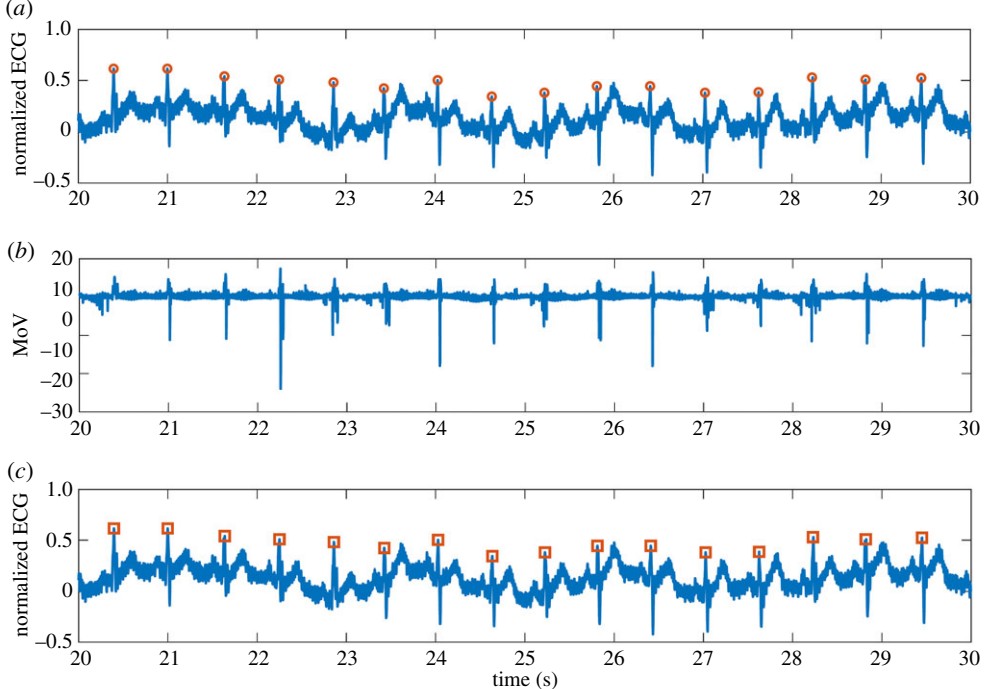

**Figure 4.** (*a*) Characteristic heartbeat of subject 122 from the MIT-BIH Arrhythmia database [8], (*b*) corresponding MoV and (*c*) ECG with the R-wave identified using our approach based on MoV.

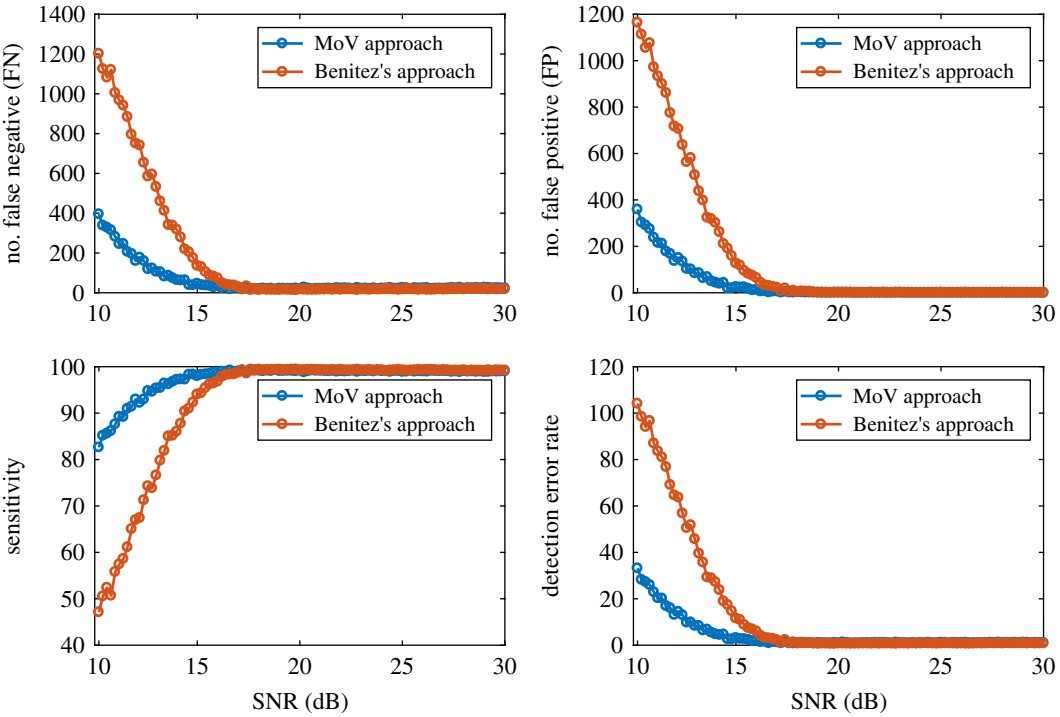

**Figure 5.** The performance of MoV versus the approach of Benitez *et al.* [16] when the original ECG signal is intentionally contaminated with different amounts of noise. The number of false negatives and false positives, sensitivity and detection error rate for both approaches are demonstrated.

## 4.2. Performance of MoV in noisy conditions

The detection of ECG signals introduces noise in the channel together with noise due to unwanted muscle activity. To examine the impact of noise on the performance of MoV, we intentionally add

**Table 1.** The R wave detected results for each individual subject from the database.

| subject ID* | manual annotation | MoV approach | failed detection | FP | FN | detection error rate (%) | SE | PPV |
|---|---|---|---|---|---|---|---|---|
| 100 | 2273 | 2273 | 0 | 0 | 0 | 0.00 | 100 | 100 |
| 101 | 1865 | 1864 | 3 | 1 | 2 | 0.16 | 99.89 | 99.95 |
| 102 | 2187 | 2186 | 1 | 0 | 1 | 0.04 | 99.95 | 100 |
| 103 | 2084 | 2082 | 2 | 0 | 2 | 0.09 | 99.90 | 100 |
| 104 | 2229 | 2230 | 3 | 2 | 1 | 0.13 | 99.96 | 99.91 |
| 105 | 2572 | 2573 | 5 | 3 | 2 | 0.19 | 99.92 | 99.88 |
| 106 | 2027 | 2027 | 0 | 0 | 0 | 0.00 | 100 | 100 |
| 107 | 2137 | 2136 | 3 | 1 | 2 | 0.14 | 99.91 | 99.95 |
| 109 | 2532 | 2531 | 1 | 0 | 1 | 0.03 | 99.96 | 100 |
| 111 | 2124 | 2125 | 1 | 1 | 0 | 0.04 | 100 | 99.95 |
| 112 | 2539 | 2539 | 0 | 0 | 0 | 0.00 | 100 | 100 |
| 113 | 1795 | 1795 | 2 | 1 | 1 | 0.11 | 99.94 | 99.94 |
| 114 | 1879 | 1878 | 3 | 1 | 2 | 0.15 | 99.89 | 99.95 |
| 115 | 1953 | 1953 | 0 | 0 | 0 | 0.00 | 100 | 100 |
| 116 | 2412 | 2411 | 1 | 0 | 1 | 0.04 | 99.96 | 100 |
| 117 | 1535 | 1534 | 1 | 0 | 1 | 0.06 | 99.93 | 100 |
| 118 | 2278 | 2279 | 1 | 1 | 0 | 0.04 | 100 | 99.96 |
| 119 | 1987 | 1987 | 0 | 0 | 0 | 0.00 | 100 | 100 |
| 121 | 1863 | 1861 | 2 | 0 | 2 | 0.10 | 99.89 | 100 |
| 122 | 2476 | 2475 | 1 | 0 | 1 | 0.04 | 99.96 | 100 |
| 123 | 1518 | 1518 | 2 | 1 | 1 | 0.13 | 99.93 | 99.93 |
| 124 | 1619 | 1619 | 0 | 0 | 0 | 0.00 | 100 | 100 |
| average | | | | | | 0.07 | 99.95 | 99.97 |

*Note that we have only used the subjects considered in Benitez's study in order to make a direct comparison.

different amounts of Gaussian white noise to the ECG signal of subject no. 100 from the MIT-BIH Arrhythmia database [8], and evaluate both MoV and the approach of Benitez et al. [16]. The approach of Benitez is a Hilbert transform-based method; and so we select this for purposes of comparison. It applies a useful Hilbert transform property that takes a zero crossing between consecutive positive and negative inflection points in the original waveform and represents it as a peak in its Hilbert transform conjugate. Therefore, in order to transform R peaks to zero crossings, the algorithm first obtains the derivative of the ECG signal. Then it applies the Hilbert transform to obtain peaks from zero crossing points. Finally, it uses instantaneous amplitude of the analytic signal to highlight R waves. In order to obtain a fair comparison, the same peak detection algorithm is applied to both approaches in a similar way. Performance is then assessed by comparison with manual annotations.

A false negative (FN) occurs when a true QRS complex identified in the corresponding manual annotation file is missed. Moreover, a false QRS detection is denoted as a false positive (FP). Sensitivity (SE) and detection error rate, respectively, are calculated using the following equations:

$$\text{sensitivity (SE)} = \frac{\text{TP}}{\text{TP} + \text{FN}} \times 100, \tag{4.1}$$

where TP denotes the numbers of true positive detections, and

$$\text{detection error rate} = \frac{\text{FP} + \text{FN}}{\text{number of heartbeats}} \times 100. \tag{4.2}$$

**Table 2.** The R-wave identification accuracy for each individual subject is calculated using the MoV approach and the Benitez approach [16].

| subject ID | SE (%) | | PPV (%) | |
|---|---|---|---|---|
| | MoV approach | the Benitez approach [16] | MoV approach | the Benitez approach [16] |
| 100 | 100 | 100 | 100 | 100 |
| 101 | 99.89 | 99.95 | 99.95 | 99.84 |
| 102 | 99.95 | 99.91 | 100 | 99.95 |
| 103 | 99.9 | 100 | 100 | 100 |
| 104 | 99.96 | 99.78 | 99.91 | 99.46 |
| 105 | 99.92 | 99.88 | 99.88 | 99.73 |
| 106 | 100 | 100 | 100 | 99.95 |
| 107 | 99.91 | 99.67 | 99.95 | 99.95 |
| 109 | 99.96 | 99.72 | 100 | 99.96 |
| 111 | 100 | 99.95 | 99.95 | 99.95 |
| 112 | 100 | 100 | 100 | 100 |
| 113 | 99.94 | 100 | 99.94 | 100 |
| 114 | 99.89 | 100 | 99.95 | 99.95 |
| 115 | 100 | 100 | 100 | 100 |
| 116 | 99.96 | 100 | 100 | 100 |
| 117 | 99.93 | 99.93 | 100 | 99.93 |
| 118 | 100 | 100 | 99.96 | 99.96 |
| 119 | 100 | 100 | 100 | 99.95 |
| 121 | 99.89 | 99.95 | 100 | 100 |
| 122 | 99.96 | 100 | 100 | 100 |
| 123 | 99.93 | 100 | 99.93 | 99.93 |
| 124 | 100 | 100 | 100 | 100 |
| average | 99.95 | 99.94 | 99.97 | 99.93 |

The performance of MoV versus the approach of Benitez *et al.* [16] is evaluated in figure 5 that shows MoV is more reliable than Benitez's approach in noisy conditions. In particular, the detection error rate of MoV is considerably lower for SNR levels worse than 18 dB.

## 4.3. Results

The results for 22 subjects obtained from the MIT-BIH Arrhythmia database are illustrated in table 1. The performance of MoV approach against the approach of Benitez *et al.* [16] is evaluated in table 2 as a number of other approaches have also been compared to the study by Benitez *et al.* [16]. Note that no noise is added in this experiment. Here, positive predictive values (PPV) respectively are calculated using the following equation,

$$PPV = \frac{TP}{TP + FP} \times 100. \tag{4.3}$$

## 5. Conclusion

MoV is introduced as a tool for the analysis of non-stationary signals. The approach is based on IF and yet is more robust to noisy conditions. As a case example, we demonstrate that for a noisy ECG signal, the loss in signal features in an IF plot is dramatic. Moreover, using the MIT-BIH Arrhythmia database, the

approach performed successfully with accurate R-wave identification, even when the original signal is contaminated with noise. The comparison between MoV and a different Hilbert transform-based approach indicates the advantage of MoV in the case of R-wave identification in noisy conditions. Our MoV approach demonstrates reduced error especially for 18 dB SNR or less. The result also suggests that MoV may be considered as an alternative method to IF or other instantaneous parameters in other non-stationary applications, particularly in noisy conditions.

Data accessibility. The program code for the analyses is at https://github.com/Dorraki/Moment-of-Velocity.git.
Authors' contributions. D.A. and B.R.D. developed the main idea; M.D. and A.F. performed the analysis and wrote the paper; D.A., A.A. and M.D. conceived the study; D.A. and A.A. supervised the study; B.R.D. assisted with the analysis, and all authors proofed the manuscript. All authors contributed to interpretation of results.
Competing interests. No competing interests.
Funding. There was no funding.
Acknowledgements. The authors acknowledge S. Messer for assistance in an earlier stage of this work.

# Appendix A

Negative values may appear in the IF waveforms, which is meaningless physically. Note that DC offsets, riding waves and abrupt changes in the signal under analysis can be the main cause of negative IF. Figure 6 shows how DC offset may potentially influence the sign of IF. Figure 6a demonstrates three signals with different DC terms, $s(t) = \sin(2\pi t)$, $\sin(2\pi t) + 0.5$ and $\sin(2\pi t) + 2$. In figure 6b, the trajectory of the signals versus their Hilbert transforms are shown. The trajectory for all the signals is a circle with the centres shifted by different amounts. Finally, the corresponding instantaneous phase and frequency diagrams are shown in figure 6c,d, respectively.

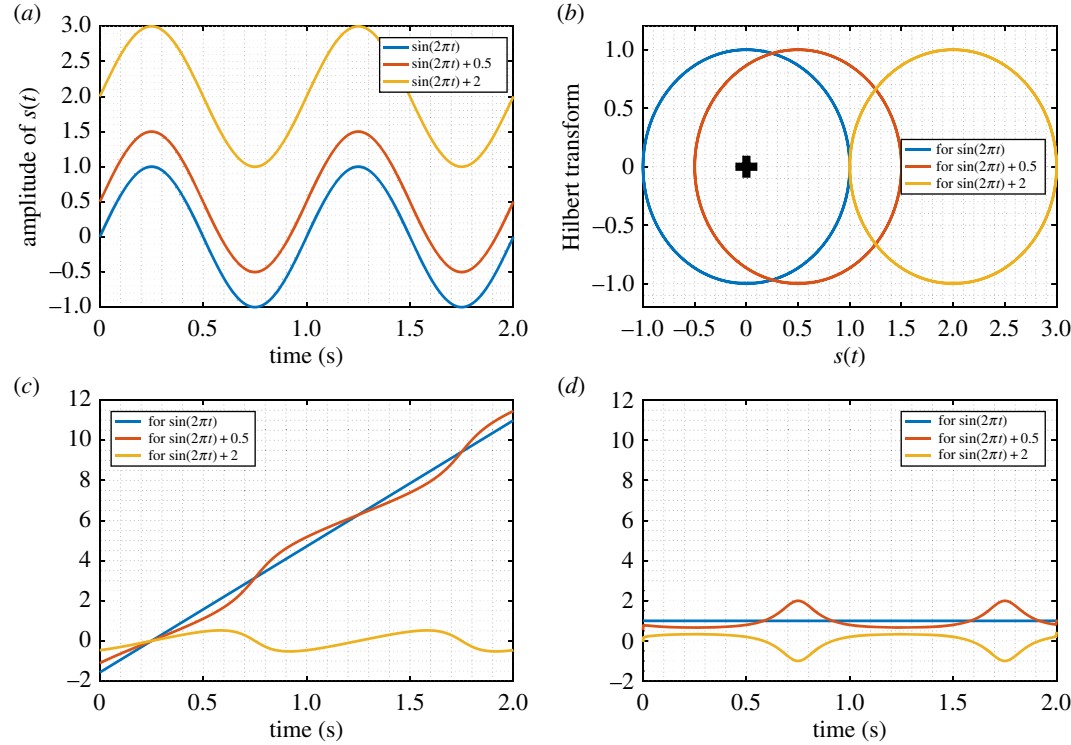

**Figure 6.** Signals with different DC term and their instantaneous parameters. (a) Signals, (b) Hilbert diagram, (c) instantaneous phase and (d) instantaneous frequency.

## A.1. No DC offset

For the case of zero DC offset, $s(t) = \sin(2t)$, the trajectory of signal and Hilbert transform is a circle with its centre exactly located on the origin; see blue circle in figure 6b. Therefore, a point moving on the circle

with a constant angular speed, $\omega$, possesses a linear instantaneous phase with constant slope; see blue line in figure 6c. The corresponding IF that is a straight line is shown in figure 6d.

## A.2. Low DC offset

In the presence of a small DC offset less than the signal amplitude, $s(t) = \sin(2\pi t) + 0.5$, the origin is still located in the Hilbert transform circle but is not located exactly on the centre of the circle; see red circle in figure 6b. Thus, for a point travelling on the circle with constant $\omega$, the phase slope is time-varying. It may be seen that from the blue curve in figure 6c that the slope is always positive, therefore, the corresponding IF possesses some positive peaks in the location of major phase variations.

## A.3. High DC offset

In this case, $s(t) = \sin(2\pi t) + 2$, the origin is located out of the Hilbert circle; see yellow circle in figure 6b. Therefore, for a moving point with constant $\omega$, the instantaneous phase possesses negative slope in each period. These negative slopes in the instantaneous phase diagram cause negative values in the corresponding IF, see yellow curve in figure 6d.

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
