## [Reviewer comments · Royal Society Open Science]

Review History

RSOS-180975.R0 (Original submission)

Review form: Reviewer 1

Is the manuscript scientifically sound in its present form?

No

Are the interpretations and conclusions justified by the results?

No

Is the language acceptable?

No

Is it clear how to access all supporting data?

Yes

Do you have any ethical concerns with this paper?

No

Have you any concerns about statistical analyses in this paper?

Yes

Recommendation?

Reject

Comments to the Author(s)

On moment of velocity (MoV) for signal analysis

The investigators introduce the concept of moment of velocity (MoV) for signal analysis. As a case study, they compare the performance of moment of velocity to a standard Hilbert transform based approach for R wave identification in human ECG signals. They found that their approach was more robust to noise.

Analysis of biosignals is an important topic. In particular, the electrocardiogram is of great interest in medicine. Its morphology and the timing of the waves can provide important information concerning cardiac health.

I am not sure how convincing the author case study is. There is just one example. Furthermore, I am unsure as to what the authors are trying to detect. In Figure 2, there are obviously a lot of spikes and randomness in the instantaneous frequency. The authors need to explain it. What do the deflections mean? Why is there a burst in the middle? Why aren't the deflections symmetric with respect to the time series? (middle panels).

The authors display the MoV in the lower panels. There are a pair of deflections that seem to coincide with the R waves of the ECG in the top panels. But so what? This is not hard to do. Adding random noise to the system and still detecting the R waves is also not so impressive. There is very limited random noise. A threshold could also be used to detect the R wave peaks, even with the random noise.

What exactly is being detected and measured? Identification means? The timing of the peak of the R wave? Because it could also mean to classify R waves of differing morphology, among other things. The authors should state that the purpose of the MoV algorithm is for R peak timing detection if that is true.

In Figure 3 there are very clean signals. So it is not surprising that the MoV finds the peaks. I would like to see the MoV find the peaks in data with large motion artifact, very irregular heartbeat, and large level of noise - noise on the order or much greater than the signal magnitude, and nonrandom noise such as spikes and chirps.

There are grammatical errors in the manuscript. Also, some descriptions seem unnecessary, such as the ECG waveform components and Figure 1.

Review form: Reviewer 2 (Jason Ralph)**Is the manuscript scientifically sound in its present form?**

Yes

Are the interpretations and conclusions justified by the results?

Yes

Is the language acceptable?

Yes

Is it clear how to access all supporting data?

Yes

Do you have any ethical concerns with this paper?

No

Have you any concerns about statistical analyses in this paper?

No

Recommendation?

Accept with minor revision (please list in comments)

Comments to the Author(s)

The manuscript describes a mathematical tool that can be used to provide a more numerically stable estimate for the instantaneous frequency of a time-dependent one-dimensional signal. The work is certainly new to me, and the methods and the results when applied to an openly accessible database are very well described and the conclusions are justified. I only have two areas where the paper could be improved. The first is that the presentation of the data and the results could use a standard method for specifying the performance. At the moment, the number of false negatives (FNs) and false positives (FPs) are given in Figure 4, but the number alone only shows that the method presented is superior to Benitez's method - no total number of examples is given. A more conventional representation, either based on the the confusion matrix or, preferably, the Receiver-Operator Characteristic (ROC) curve should be given. In addition, it appears that the results may only relate to one of the MIT/BITH examples present in the database. It would be better to be more explicit about which of the examples have been used and give more than one example. For instance, are the results typical and in what circumstances/for which examples does the method fail?

If these comments are addressed suitably, I would recommend publication.

Review form: Reviewer 3 (David Halliday)

Is the manuscript scientifically sound in its present form?

No

Are the interpretations and conclusions justified by the results?

Yes

Is the language acceptable?

Yes

Is it clear how to access all supporting data?

Yes

Do you have any ethical concerns with this paper?

No

Have you any concerns about statistical analyses in this paper?

No

Recommendation?

Major revision is needed (please make suggestions in comments)

Comments to the Author(s)

Comments to authors.

This is an interesting manuscript which presents an alternative method for calculating instantaneous frequency based on a quaternion type representation, which appears to have improved performance against noise when applied to ECG records.

I found the development of the MoV framework a little terse – it could benefit from an expanded treatment. Points to address are

1. Page 3 “MoV has a linear relationship with IF”. It would be useful to elucidate this in more detail, what exactly is meant by “linear relationship”
2. The material in equation 3.2 – 3.6 needs a more systematic treatment and introduction. Thus why is this model appropriate for neurophysiological time series?
3. The terms in equations 3.2 need defined and introduced, and a link provided to the analytic signal representation, where, for example, the concept of mass does not make sense?
4. Equation 3.3 needs completed, include a LHS term for clarity and define your unit vectors i, j, k .
5. It would be useful to comment on the broader context of the MoV measure. There is considerable interest in instantaneous phase and phase coupling measures, but instantaneous phase is only mentioned briefly in the Appendix. The relationship between MoV and instantaneous phase should be explained in the methods section.
6. An alternative approach to calculating IF is the use of complex wavelets, again a comment on how MoV compares to complex wavelets would help provide a clearer context for the work.

The other structural issue I wondered about was the presentation of the results. A single ECG example is presented, taken from an online data base, which presumably has more entries. It would make a more convincing case if the MoV could be applied across a number of data sets to achieve a stronger validation.

Minor

1. Page 2. “Therefore, they are useless for a signal” I would say “not appropriate” instead of “useless”
2. Page 3, section 3, Typo “dominator”
3. Page 5. “An investigation of ECG signals based on...” Poor grammar here.
4. Page 6, Figure 2. No units are included for MoV plots.

David Halliday,
University of York.

Decision letter (RSOS-180975.R0)

27-Sep-2018

Dear Mr Dorraki:

Manuscript ID RSOS-180975 entitled "On moment of velocity (MoV) for signal analysis" which you submitted to Royal Society Open Science, has been reviewed. The comments from reviewers are included at the bottom of this letter.

In view of the criticisms of the reviewers, the manuscript has been rejected in its current form. However, a new manuscript may be submitted which takes into consideration these comments.

Please note that resubmitting your manuscript does not guarantee eventual acceptance, and that your resubmission will be subject to peer review before a decision is made.

Your resubmitted manuscript should be submitted by 27-Mar-2019. If you are unable to submit by this date please contact the Editorial Office.

Please note that Royal Society Open Science will introduce article processing charges for all new submissions received from 1 January 2018. Charges will also apply to papers transferred to Royal Society Open Science from other Royal Society Publishing journals, as well as papers submitted as part of our collaboration with the Royal Society of Chemistry (<http://rsos.royalsocietypublishing.org/chemistry>). If your manuscript is submitted and accepted for publication after 1 Jan 2018, you will be asked to pay the article processing charge, unless you request a waiver and this is approved by Royal Society Publishing. You can find out more about the charges at <http://rsos.royalsocietypublishing.org/page/charges>. Should you have any queries, please contact openscience@royalsociety.org.

on behalf of Prof. R. Kerry Rowe (Subject Editor)
openscience@royalsociety.org

Associate Editor Comments:

The referees have identified a number of concerns with the paper, though also a number of positives in the study. Given the nature of the changes required by the referees, and to ensure you have sufficient time to complete these changes, the paper will be rejected in its current form

but a resubmission would be welcomed by the journal, taking into consideration the concerns raised by the referees.

Reviewers' Comments to Author:

Reviewer: 1

Comments to the Author(s)

On moment of velocity (MoV) for signal analysis

The investigators introduce the concept of moment of velocity (MoV) for signal analysis. As a case study, they compare the performance of moment of velocity to a standard Hilbert transform based approach for R wave identification in human ECG signals. They found that their approach was more robust to noise.

Analysis of biosignals is an important topic. In particular, the electrocardiogram is of great interest in medicine. Its morphology and the timing of the waves can provide important information concerning cardiac health.

I am not sure how convincing the author case study is. There is just one example. Furthermore, I am unsure as to what the authors are trying to detect. In Figure 2, there are obviously a lot of spikes and randomness in the instantaneous frequency. The authors need to explain it. What do the deflections mean? Why is there a burst in the middle? Why aren't the deflections symmetric with respect to the time series? (middle panels).

The authors display the MoV in the lower panels. There are a pair of deflections that seem to coincide with the R waves of the ECG in the top panels. But so what? This is not hard to do. Adding random noise to the system and still detecting the R waves is also not so impressive. There is very limited random noise. A threshold could also be used to detect the R wave peaks, even with the random noise.

What exactly is being detected and measured? Identification means? The timing of the peak of the R wave? Because it could also mean to classify R waves of differing morphology, among other things. The authors should state that the purpose of the MoV algorithm is for R peak timing detection if that is true.

In Figure 3 there are very clean signals. So it is not surprising that the MoV finds the peaks. I would like to see the MoV find the peaks in data with large motion artifact, very irregular heartbeat, and large level of noise - noise on the order or much greater than the signal magnitude, and nonrandom noise such as spikes and chirps.

There are grammatical errors in the manuscript. Also, some descriptions seem unnecessary, such as the ECG waveform components and Figure 1.

Reviewer: 2

Comments to the Author(s)

The manuscript describes a mathematical tool that can be used to provide a more numerically stable estimate for the instantaneous frequency of a time-dependent one-dimensional signal. The work is certainly new to me, and the methods and the results when applied to an openly accessible database are very well described and the conclusions are justified. I only have two areas where the paper could be improved. The first is that the presentation of the data and the results could use a standard method for specifying the performance. At the moment, the number

of false negatives (FNs) and false positives (FPs) are given in Figure 4, but the number alone only shows that the method presented is superior to Benitez's method - no total number of examples is given. A more conventional representation, either based on the the confusion matrix or, preferably, the Receiver-Operator Characteristic (ROC) curve should be given. In addition, it appears that the results may only relate to one of the MIT/BITH examples present in the database. It would be better to be more explicit about which of the examples have been used and give more than one example. For instance, are the results typical and in what circumstances/for which examples does the method fail?

If these comments are addressed suitably, I would recommend publication.

Reviewer: 3

Comments to the Author(s)

Comments to authors.

This is an interesting manuscript which presents an alternative method for calculating instantaneous frequency based on a quaternion type representation, which appears to have improved performance against noise when applied to ECG records.

I found the development of the MoV framework a little terse – it could benefit from an expanded treatment. Points to address are

1. Page 3 “MoV has a linear relationship with IF”. It would be useful to elucidate this in more detail, what exactly is meant by “linear relationship”
2. The material in equation 3.2 – 3.6 needs a more systematic treatment and introduction. Thus why is this model appropriate for neurophysiological time series?
3. The terms in equations 3.2 need defined and introduced, and a link provided to the analytic signal representation, where, for example, the concept of mass does not make sense?
4. Equation 3.3 needs completed, include a LHS term for clarity and define your unit vectors i, j, k .
5. It would be useful to comment on the broader context of the MoV measure. There is considerable interest in instantaneous phase and phase coupling measures, but instantaneous phase is only mentioned briefly in the Appendix. The relationship between MoV and instantaneous phase should be explained in the methods section.
6. An alternative approach to calculating IF is the use of complex wavelets, again a comment on how MoV compares to complex wavelets would help provide a clearer context for the work.

The other structural issue I wondered about was the presentation of the results. A single ECG example is presented, taken from an online data base, which presumably has more entries. It would make a more convincing case if the MoV could be applied across a number of data sets to achieve a stronger validation.

Minor

1. Page 2. “Therefore, they are useless for a signal” I would say “not appropriate” instead of “useless”
2. Page 3, section 3, Typo “dominator”
3. Page 5. “An investigation of ECG signals based on...” Poor grammar here.
4. Page 6, Figure 2. No units are included for MoV plots.

David Halliday,
University of York.

Author's Response to Decision Letter for (RSOS-180975.R0)

See Appendix A.

RSOS-182001.R0

Review form: Reviewer 1

Is the manuscript scientifically sound in its present form?

No

Are the interpretations and conclusions justified by the results?

No

Is the language acceptable?

No

Is it clear how to access all supporting data?

No

Do you have any ethical concerns with this paper?

No

Have you any concerns about statistical analyses in this paper?

No

Recommendation?

Major revision is needed (please make suggestions in comments)

Comments to the Author(s)

The manuscript should be refined for English grammatical structure and phraseology. The manuscript should be polished by an English language service (note in marked-up copy text where changes are made). Details of author-pays services can be found, for example, at:

<http://www.proof-reading-service.com/>, <http://www.aje.com/us/>,

<http://webshop.elsevier.com/languageediting/>, <http://americanmanuscripteditors.com/>,

<https://www.servicescape.com/>, <https://www.capstoneediting.com.au/>, or

<http://eworldediting.com>. Examples –

It is investigated three possible conditions:

Because IF is derivation of phase based eq. 2.6,

Authors have created 5 figures and 2 tables. I think they are allowed another figure, in which case they should create another like Figure 3, with different and more difficult data.

Explanation of Figure 5 needs remedy. Explain why red signal has similar instantaneous phase to blue signal, but yellow signal is flat (panel c). Explain why red versus yellow signals are inverted in panel d.

Review form: Reviewer 3 (David Halliday)

Is the manuscript scientifically sound in its present form?

Yes

Are the interpretations and conclusions justified by the results?

Yes

Is the language acceptable?

Yes

Is it clear how to access all supporting data?

Yes

Do you have any ethical concerns with this paper?

No

Have you any concerns about statistical analyses in this paper?

No

Recommendation?

Accept as is

Comments to the Author(s)

This is reviewer #3 from before. I am happy with the changes made. I have one minor point - in the conclusions the point is made that the performance of the MOV method is better than the comparison below 18dB SNR. It would be useful to indicate if this is within the range typically seen for ECG recordings.

Decision letter (RSOS-182001.R0)

18-Dec-2018

Dear Mr Dorraki,

The Subject Editor assigned to your paper ("On moment of velocity (MoV) for signal analysis") has now received comments from reviewers. We would like you to revise your paper in accordance with the referee and Associate Editor suggestions which can be found below (not including confidential reports to the Editor). Please note this decision does not guarantee eventual acceptance.

Please submit a copy of your revised paper before 10-Jan-2019. Please note that the revision deadline will expire at 00.00am on this date. If we do not hear from you within this time then it will be assumed that the paper has been withdrawn. In exceptional circumstances, extensions may be possible if agreed with the Editorial Office in advance. We do not allow multiple rounds of revision so we urge you to make every effort to fully address all of the comments at this stage. If deemed necessary by the Editors, your manuscript will be sent back to one or more of the original reviewers for assessment. If the original reviewers are not available we may invite new reviewers.

When submitting your revised manuscript, you must respond to the comments made by the referees and upload a file "Response to Referees" in "Section 6 - File Upload". Please use this to document how you have responded to each of the comments, and the adjustments you have made. In order to expedite the processing of the revised manuscript, please be as specific as possible in your response.

- Ethics statement

- Data accessibility

If you wish to submit your supporting data or code to Dryad (<http://datadryad.org/>), or modify your current submission to dryad, please use the following link:
<http://datadryad.org/submit?journalID=RSOS&manu=RSOS-182001>

- Competing interests

- Authors' contributions

- Acknowledgements

- Funding statement

Please note that Royal Society Open Science charge article processing charges for all new submissions that are accepted for publication. Charges will also apply to papers transferred to Royal Society Open Science from other Royal Society Publishing journals, as well as papers submitted as part of our collaboration with the Royal Society of Chemistry (<http://rsos.royalsocietypublishing.org/chemistry>). If your manuscript is newly submitted and subsequently accepted for publication, you will be asked to pay the article processing charge, unless you request a waiver and this is approved by Royal Society Publishing. You can find out more about the charges at <http://rsos.royalsocietypublishing.org/page/charges>. Should you have any queries, please contact openscience@royalsociety.org.

on behalf of Prof R. Kerry Rowe (Subject Editor)
openscience@royalsociety.org

Editor comments:

Reviewer 1 continues to have some concerns with the paper, including the quality of the written English. With this in mind, we'd like you to incorporate or rebut the concerns of that referee and also seek advice from a professional English Language editing service (for example, from the list at <https://royalsociety.org/journals/authors/language-polishing/>) before resubmitting. You should provide evidence of having sought advice, and the paper will be returned to you if you do not do so.

Reviewer comments to Author:

Reviewer: 1

Comments to the Author(s)

The manuscript should be refined for English grammatical structure and phraseology. The manuscript should be polished by an English language service (note in marked-up copy text where changes are made). Details of author-pays services can be found, for example, at: <http://www.proof-reading-service.com/>, <http://www.aje.com/us/>,

<http://webshop.elsevier.com/languageediting/>, <http://americanmanuscripteditors.com/>,
<https://www.servicescape.com/>, <https://www.capstoneediting.com.au/>, or
<http://eworldediting.com>. Examples –

It is investigated three possible conditions:

Because IF is derivation of phase based eq. 2.6,

Authors have created 5 figures and 2 tables. I think they are allowed another figure, in which case they should create another like Figure 3, with different and more difficult data.

Explanation of Figure 5 needs remedy. Explain why red signal has similar instantaneous phase to blue signal, but yellow signal is flat (panel c). Explain why red versus yellow signals are inverted in panel d.

Reviewer: 3

Comments to the Author(s)

This is reviewer #3 from before. I am happy with the changes made. I have one minor point - in the conclusions the point is made that the performance of the MOV method is better than the comparison below 18dB SNR. It would be useful to indicate if this is within the range typically seen for ECG recordings.

Author's Response to Decision Letter for (RSOS-182001.R0)

See Appendix B.

RSOS-182001.R1 (Revision)

Review form: Reviewer 1

Is the manuscript scientifically sound in its present form?

Yes

Are the interpretations and conclusions justified by the results?

Yes

Is the language acceptable?

No

Is it clear how to access all supporting data?

Yes

Do you have any ethical concerns with this paper?

No

Have you any concerns about statistical analyses in this paper?

No

Recommendation?

Major revision is needed (please make suggestions in comments)

Comments to the Author(s)

On moment of velocity (MoV) for signal analysis

There are still typos/corrections needed. Examples -

Fig. 6 shows that how

transform is a circle that its centre is exactly

The corresponding IF that is a straight line is show in Fig.

These negative slopes in instantaneous phase diagram cause

Here is the blurb I would use at my own journal at this point (but the EIC here may be easier) -
 Editor-In-Chief: The manuscript should be refined for English grammatical structure and phraseology. The manuscript should be polished by an English language service (note in marked-up copy text where changes are made). Details of author-pays services can be found, for example, at: <http://www.sciencemanager.com/>, <http://www.proof-reading-service.com/>, <http://www.aje.com/us/>, <http://webshop.elsevier.com/languageediting/>, <http://americanmanuscripteditors.com/>, <https://www.servicescape.com/>, <https://www.capstoneediting.com.au/>, or <http://eworlddediting.com>. If not done, manuscript should be withdrawn or shall be rejected on next round.

You refer to a brown trace. I am not sure that is clear. I see dark red and blue and dark yellow. Both the dark red and dark yellow could be interpreted as brown.

Decision letter (RSOS-182001.R1)

18-Feb-2019

Dear Mr Dorraiki:

Manuscript ID RSOS-182001.R1 entitled "On moment of velocity (MoV) for signal analysis" which you submitted to Royal Society Open Science, has been reviewed. The comments of the reviewer(s) are included at the bottom of this letter.

Please submit a copy of your revised paper before 13-Mar-2019. Please note that the revision deadline will expire at 00.00am on this date. If we do not hear from you within this time then it will be assumed that the paper has been withdrawn. In exceptional circumstances, extensions may be possible if agreed with the Editorial Office in advance. We do not allow multiple rounds of revision so we urge you to make every effort to fully address all of the comments at this stage. If deemed necessary by the Editors, your manuscript will be sent back to one or more of the original reviewers for assessment. If the original reviewers are not available we may invite new reviewers.

To revise your manuscript, log into <http://mc.manuscriptcentral.com/rsos> and enter your Author Centre, where you will find your manuscript title listed under "Manuscripts with Decisions." Under "Actions," click on "Create a Revision." Your manuscript number has been

appended to denote a revision. Revise your manuscript and upload a new version through your Author Centre.

- Ethics statement

- Data accessibility

- Competing interests

- Authors' contributions

- Acknowledgements

- Funding statement

on behalf of Prof R. Kerry Rowe (Subject Editor)
 openscience@royalsociety.org

Editor comments:

The reviewer is largely persuaded that the paper is scientifically sound and ready for publication on that score; however, concerns remain regarding the language. With this in mind, we'd like you to please seek the advice of a language polishing service (examples may be found at <https://royalsociety.org/journals/authors/language-polishing/>), and once you have done so, resubmit and provide evidence of having sought advice (an editorial certificate, for instance). Once this has been completed, the paper will be accepted for publication.

Reviewer comments to Author:

Reviewer: 1

Comments to the Author(s)

On moment of velocity (MoV) for signal analysis

There are still typos/corrections needed. Examples -

Fig. 6 shows that how

transform is a circle that its centre is exactly

The corresponding IF that is a straight line is show in Fig.

These negative slopes in instantaneous phase diagram cause

Here is the blurb I would use at my own journal at this point (but the EIC here may be easier) -
 Editor-In-Chief: The manuscript should be refined for English grammatical structure and phraseology. The manuscript should be polished by an English language service (note in marked-up copy text where changes are made). Details of author-pays services can be found, for example, at: <http://www.sciencemanager.com/>, <http://www.proof-reading-service.com/>, <http://www.aje.com/us/>, <http://webshop.elsevier.com/languageediting/>, <http://americanmanuscripteditors.com/>, <https://www.servicescape.com/>, <https://www.capstoneediting.com.au/>, or <http://eworlddediting.com>. If not done, manuscript should be withdrawn or shall be rejected on next round.

You refer to a brown trace. I am not sure that is clear. I see dark red and blue and dark yellow. Both the dark red and dark yellow could be interpreted as brown.

Author's Response to Decision Letter for (RSOS-182001.R1)

See Appendix C.

Decision letter (RSOS-182001.R2)

27-Feb-2019

Dear Mr Dorraki,

I am pleased to inform you that your manuscript entitled "On moment of velocity (MoV) for signal analysis" is now accepted for publication in Royal Society Open Science.

on behalf of Professor R. Kerry Rowe (Subject Editor)
openscience@royalsociety.org

Appendix A

Manuscript ID: RSOS-180975

Title: On moment of velocity (MoV) for signal analysis

Author: Dorraki *et al*

To: Prof. R. Kerry Rowe, Editor

Dear Kerry,

Re: Reply to reviewers on RSOS-180975

Many thanks for the valuable comments on our paper RSOS-180975. We have pleasure in attaching an updated manuscript and our point-by-point response to the comments is given below.

Best regards,

Mohsen Dorraki *et al.*

Reviewer#1: I am not sure how convincing the author case study is. There is just one example.

Author reply: We agree.

Author action: We now increase the number of examples to 22 time series. Table 1 containing related results such as false positive, false negative, detection error rate, sensitivity, and positive predictive values is added to the manuscript. Moreover, in Table 2 the R-wave identification accuracy for each individual subject is calculated using the MoV and the Benitez approaches.

Table 1: The R wave detected result for each individual subject from the database

Subject ID	Manual annotation	MoV approach	Failed detection	FP	FN	Detection error rate (%)	SE	PPV
100	2273	2273	0	0	0	0	100	100
101	1865	1864	3	1	2	0.1608	99.89	99.95
102	2187	2186	1	0	1	0.0457	99.95	100
103	2084	2082	2	0	2	0.0959	99.9	100
104	2229	2230	3	2	1	0.1345	99.96	99.91
105	2572	2573	5	3	2	0.1944	99.92	99.88
106	2027	2027	0	0	0	0	100	100
107	2137	2136	3	1	2	0.1403	99.91	99.95
109	2532	2531	1	0	1	0.0394	99.96	100
111	2124	2125	1	1	0	0.0470	100	99.95
112	2539	2539	0	0	0	0	100	100
113	1795	1795	2	1	1	0.1114	99.94	99.94
114	1879	1878	3	1	2	0.1596	99.89	99.95
115	1953	1953	0	0	0	0	100	100
116	2412	2411	1	0	1	0.0414	99.96	100
117	1535	1534	1	0	1	0.0651	99.93	100
118	2278	2279	1	1	0	0.0438	100	99.96
119	1987	1987	0	0	0	0	100	100
121	1863	1861	2	0	2	0.1073	99.89	100
122	2476	2475	1	0	1	0.0403	99.96	100
123	1518	1518	2	1	1	0.1317	99.93	99.93
124	1619	1619	0	0	0	0	100	100
Average						0.07088	99.9540	99.9736

Table 2: The R-wave identification accuracy for each individual subject is calculated using the MoV approach and the Hilbert approach.

Subject ID	SE (%)		PPV (%)	
	MoV approach	The Hilbert approach	MoV approach	The Hilbert approach
100	100	100	100	100
101	99.89	99.95	99.95	99.84
102	99.95	99.91	100	99.95
103	99.9	100	100	100
104	99.96	99.78	99.91	99.46
105	99.92	99.88	99.88	99.73
106	100	100	100	99.95
107	99.91	99.67	99.95	99.95
109	99.96	99.72	100	99.96
111	100	99.95	99.95	99.95
112	100	100	100	100
113	99.94	100	99.94	100
114	99.89	100	99.95	99.95
115	100	100	100	100
116	99.96	100	100	100
117	99.93	99.93	100	99.93
118	100	100	99.96	99.96
119	100	100	100	99.95
121	99.89	99.95	100	100
122	99.96	100	100	100
123	99.93	100	99.93	99.93
124	100	100	100	100
124	99.95	99.94	99.97	99.93

Reviewer#1: Furthermore, I am unsure as to what the authors are trying to detect. In Figure 2, there are obviously a lot of spikes and randomness in the instantaneous frequency. The authors need to explain it. What do the deflections mean? Why is there a burst in the middle? Why aren't the deflections symmetric with respect to the time series? (middle panels).

Author reply: We did in fact discuss this point in the second paragraph of introduction. We argue in this paragraph that the application of IF is limited for multicomponent signals, because instantaneous frequency (IF) normally is a fluctuating waveform, which is sensitive to the amplitudes of the frequency components. Therefore, the interpretation of spectral information is often severely constrained by the nonlinear nature of the instantaneous frequency. We highlighted that instantaneous frequency as a function of time can possess large spikes and negative values. In Appendix in Fig. 5 (d) we showed how a DC offset in the original signal leads to spikes and negative values in the IF.

In Fig. 2, we directly compare IF and MoV waveforms to show that IF contains frequency information of the signal under analysis, the information can be seen more clearly when we remove the denominator of formula.

Author action: As requested by reviewer, we clarify this further by adding to the end of third paragraph in section 4(a): "Considering Fig. 2, although, the IF waveforms possess the frequency information of the ECG signal under analysis, the information is cluttered with large spikes and negative values. The influence of DC offset, riding waves, and abrupt changes of signal are the main causes of these large spikes and negative IF values. The use of MoV can overcome this problem by removing the numerator that can be very small in the original formula."

Reviewer#1: The authors display the MoV in the lower panels. There are a pair of deflections that seem to coincide with the R waves of the ECG in the top panels. But so what? This is not hard to do. Adding random noise to the system and still detecting the R waves is also not so impressive. There is very limited random noise. A threshold could also be used to detect the R wave peaks, even with the random noise.

Author reply: In the figure, the objective is to demonstrate that how robust MoV is to noise in comparison to IF.

Author action: In order to illustrate the performance of the MoV approach we now apply the method on a group of 22 signals. The related results are provided in Tables 1 and 2.

Reviewer#1: What exactly is being detected and measured? Identification means? The timing of the peak of the R wave? Because it could also mean to classify R waves of differing morphology, among other things. The authors should state that the purpose of the MoV algorithm is for R peak timing detection if that is true.

Author reply: The particular scope of this paper is proposing MoV as a signal processing tool that can be used in many practical applications. The timing of the peak of R wave can potentially be extended as a MoV application in future studies; however, our scope is focused on investigating ECG signal as a case example to detect the exact location of R peak.

Author action: No action.

Reviewer#1: In Figure 3 there are very clean signals. So it is not surprising that the MoV finds the peaks. I would like to see the MoV find the peaks in data with large motion artifact, very irregular heartbeat, and large level of noise - noise on the order or much greater than the signal magnitude, and nonrandom noise such as spikes and chirps.

Author reply: We agree.

Author action: We now show the performance of MoV on irregular heartbeats containing artifacts and noise in Fig. 3. Moreover, we show the result for a collection of 22 ECG signals with various shapes.

Reviewer#1: There are grammatical errors in the manuscript. Also, some descriptions seem unnecessary, such as the ECG waveform components and Figure 1.

Author reply: We agree.

Author action: We now remove Figure 1 and related additional descriptions. The grammatical errors are now corrected.

Reviewer#2: I only have two areas where the paper could be improved. The first is that the presentation of the data and the results could use a standard method for specifying the performance. At the moment, the number of false negatives (FNs) and false positives (FPs) are given in Figure 4, but the number alone only shows that the method presented is superior to Benitez's method - no total number of examples is given. A more conventional representation, either based on the the confusion matrix or, preferably, the Receiver-Operator Characteristic (ROC) curve should be given.

Author reply: We agree.

Author action: We now add Table 1 so that the basic terms in the confusion matrix are indicated in it.

Reviewer#2: In addition, it appears that the results may only relate to one of the MIT/BITH examples present in the database. It would be better to be more explicit about which of the examples have been used and give more than one example. For instance, are the results typical and in what circumstances/for which examples does the method fail?

Author reply: We agree.

Author action: We now increase the number of examples to 22 time series. Table 1 containing related results such as false positive, false negative, detection error rate, sensitivity, and positive predictive values are added to the manuscript. Moreover, in Table 2 the R-wave identification accuracy for each individual subject is calculated using the MoV and the Benitez approaches.

Reviewer#3: Page 3 “MoV has a linear relationship with IF”. It would be useful to elucidate this in more detail, what exactly is meant by “linear relationship”.

Author reply: We agree that this was not appropriate.

Author action: We now remove the term “linear relationship.”

Reviewer#3: The material in equation 3.2 – 3.6 needs a more systematic treatment and introduction. Thus why is this model appropriate for neurophysiological time series?

Author reply: We defined the MoV as IF without the dominator included, and further the descriptions in Eq. 3.2 to 3.6 justify the physical basis of MoV using an angular momentum concept. The main reason this is appropriate for ECGs is that MoV is a convenient tool for analysis of non-stationary signals. We show that MoV significantly outperforms IF that is used for non-stationary signals as well, especially under noisy conditions.

In addition to describing MoV properties, we did in fact discuss the properties of ECG time series in the Section 4, as follows: “ECG signals are transient, time varying, and non-stationary. The fact that ECG signals are non-stationary imposes a number of limitations on using basic time and frequency analyses. Most time and frequency analyses are established on the assumption that signals are stationary or locally stationary. Therefore, these methods cannot represent all the characteristics of ECG signals.”

Author action: We have now extended the paragraph to further clarify the concept.

Reviewer#3: The terms in equations 3.2 need defined and introduced, and a link provided to the analytic signal representation, where, for example, the concept of mass does not make sense? Equation 3.3 needs completed, include a LHS term for clarity and define your unit vectors i, j, k .

Author reply: We agree

Author action: We clarify this further by rewriting Section 3:

“The initial concept of MoV is derived by analogy from angular momentum in the field of particle dynamics. For an analytic signal, the imaginary part is the Hilbert transform of its real part that is the original signal. Therefore, the original signal is always orthogonal to the Hilbert transform of the waveform. Similarly, the angular momentum vector is perpendicular to the position vector. The angular momentum coordinate

system is shown in Fig. 1 (a). As an example, a signal and the Hilbert transform analytic plane is demonstrated in Fig. 1 (b) for the case when $x(t) = \sin(t)$.

Fig. 1 (a) Angular momentum coordinate system, and (b) signal and the Hilbert transform analytic plane for $x(t) = \sin(t)$

The angular momentum for a moving particle of unit mass is defined as

$$L = r \times v, \quad (1)$$

where r is the position vector of particle and v is particle velocity. If the vectors r and v are resolved into components and the cross-product formula is applied we obtain,

$$L = \begin{vmatrix} \mathbf{i} & \mathbf{j} & \mathbf{k} \\ r_x & r_y & r_z \\ v_x & v_y & v_z \end{vmatrix} = (r_z v_y - r_y v_z)\mathbf{i} + (r_x v_z - r_z v_x)\mathbf{j} + (r_x v_y - r_y v_x)\mathbf{k}, \quad (2)$$

where \mathbf{i} , \mathbf{j} , \mathbf{k} , are the standard unit vectors, and indices show the x , y and z components of the vectors. Assuming the particle moves in the x - y plane, the angular momentum is perpendicular to the x - y plane and Equation 2 is reduced to only its z components defined by the scalar

$$\begin{aligned} L_z &= r_x v_y - r_y v_x \\ &= r_x \frac{dr_y}{dt} - r_y \frac{dr_x}{dt}. \end{aligned} \quad (3)$$

In terms of signal and the Hilbert transform analytic plan, we replace r_x , r_y , and L_z in Equation 3 by $x(t)$, $H(t)$ and MoV, respectively. Therefore, MoV is defined as:"

$$\text{MoV} = x(t) \frac{dH[x(t)]}{dt} - H[x(t)] \frac{dx(t)}{dt}. \quad (3)$$

Reviewer#3: It would be useful to comment on the broader context of the MoV measure. There is considerable interest in instantaneous phase and phase coupling measures, but instantaneous phase is only mentioned briefly in the Appendix. The relationship between MoV and instantaneous phase should be explained in the methods section.

Author reply: We agree

Author action: The instantaneous phase equation and its relationship with MoV is added now to the manuscript.

Reviewer#3: An alternative approach to calculating IF is the use of complex wavelets, again a comment on how MoV compares to complex wavelets would help provide a clearer context for the work.

Author reply: We thank the reviewer for this helpful comment. We have employed the IF obtain by employing complex wavelets and the results outperform IF obtained from the Hilbert transform. However, the MoV still yields superior outcomes, especially in noisy conditions.

Author action: We have added a new panel in Fig 2 showing IF obtained from the complex wavelet approach. Therefore, the explanations are changed now to:

“Figure 2 illustrates ECG, corresponding instantaneous frequency, and MoV curves for a patient from physionet database [8]. Here, IF is obtained from two different approaches, the Hilbert transform and complex wavelet. The particular complex wavelet method we used is based on the Morse wavelet [23, 24]. Additionally, 20 dB SNR noise is intentionally added to the signal and the corresponding IF and MoV waveforms are shown. In this stage, in order to directly compare IF and MoV, no filtering or pre-processing steps are carried out on the ECG.

Considering Fig. 2, although, the IF waveforms possess the frequency information of the ECG signal under analysis, the information is cluttered with large spikes and negative values, particularly in IF obtained from the Hilbert transform. The influence of DC offset, riding waves, and abrupt changes of signal are the main causes of these large spikes and negative IF values. The use of MoV can overcome this problem by removing the numerator that can be very small in the original formula.”

Fig. 2, For qualitative comparison of IF and MoV, (a) the characteristic heartbeat, corresponding (b) instantaneous frequency obtained from the Hilbert transform, (c) instantaneous frequency obtained from complex wavelet, and (d) MoV are shown for a normal patient. On the right, we have added 20 dB SNR Gaussian white noise to the ECG in order to illustrate the affect of noise.

Reviewer#3: The other structural issue I wondered about was the presentation of the results. A single ECG example is presented, taken from an online data base, which presumably has more entries. It would make a more convincing case if the MoV could be applied across a number of data sets to achieve a stronger validation.

Author reply: We agree.

Author action: We now increase the number of examples to 22 time series. Table 1 contains related results such as false positive, false negative, detection error rate, sensitivity, and positive predictive values, which are added to the manuscript. Moreover, in Table 2 the R-wave identification accuracy for each individual subject is calculated using MoV and the Benitez method.

Reviewer#3: Minor

1. Page 2. "Therefore, they are useless for a signal" I would say "not appropriate" instead of "useless"
2. Page 3, section 3, Typo "dominator"
3. Page 5. "An investigation of ECG signals based on..." Poor grammar here.
4. Page 6, Figure 2. No units are included for MoV plots.

Author reply: We agree.

Author action: Done.

Appendix B

Manuscript ID: RSOS-182001

Title: On moment of velocity (MoV) for signal analysis

Author: Dorraki *et al*

To: Prof. R. Kerry Rowe, Editor

Dear Kerry,

Re: Reply to reviewers on RSOS-182001

Many thanks for the valuable comments on our paper RSOS-182001. We have pleasure in attaching an updated manuscript and our point-by-point response to the comments is given below.

Best regards,

Mohsen Dorraki *et al*.

Editor comments: Reviewer 1 continues to have some concerns with the paper, including the quality of the written English.

Author reply: Thank you.

Author action: The manuscript has been proof read and corrected by native English speakers

Reviewer#1: The manuscript should be refined for English grammatical structure and phraseology.

Author reply: Thank you.

Author action: The manuscript has been proof read and corrected by native English speakers

Reviewer#1: Authors have created 5 figures and 2 tables. I think they are allowed another figure, in which case they should create another like Figure 3, with different and more difficult data.

Author reply: We agree.

Author action: We have now added one more figure illustrating patient No. 104's heartbeat characteristics, corresponding MoV, and detected R waves.

Figure 3: (a) Characteristic heartbeat of subject 104 from the MIT/BIH arrhythmia database [8], (b) Corresponding MoV, and (c) ECG tracing with the R-wave identified using the proposed approach.

Moreover, the last paragraph in the Section 4(a) now reads: “Fig. 3.a and Fig. 4.a demonstrate two normalized heartbeat characteristics of 10 seconds from patient 104 and 122 with manually annotated R waves. The corresponding moment of velocity curves are shown in Fig. 3.b and Fig. 4.b. The R waves detected using the algorithm are demonstrated in the lower panels, and they correctly match the manual annotations.”

Reviewer#1: Explanation of Figure 5 needs remedy. Explain why red signal has similar instantaneous phase to blue signal, but yellow signal is flat (panel c). Explain why red versus yellow signals are inverted in panel d.

Author reply: We agree.

Author action: We clarify this further by rewriting Appendix I:

“Negative values may appear in the IF waveforms, which is meaningless physically. DC offsets, riding waves, and abrupt changes in the signal under analysis can be the main cause of negative IF. Fig. 6 shows that how DC offset may potentially influence the sign of instantaneous frequency. Fig. 6(a) demonstrates three signals with different DC term, $s(t)=\sin(2\pi t)$, $\sin(2\pi t)+0.5$, and $\sin(2\pi t)+2$. In Fig. 6(b) the trajectory of the signals vs their Hilbert transforms are shown. The trajectory for all the signals is a circle with different centres. Finally, the corresponding instantaneous phase and frequency diagrams are shown in Fig. 6(c) and 6(d) respectively.

Figure 6: Signals with different DC term and their instantaneous parameters.

(i) No DC offset

In case the DC offset is zero, $s(t)=\sin(2\pi t)$, the trajectory of signal and Hilbert transform is a circle that its centre is exactly located on the origin (blue circle in Fig. 6(b)). Therefore, a point moving on the circle with a constant angular speed (ω) possesses a linear instantaneous phase with constant slope (blue line in Fig. 6(c)). The corresponding IF that is a straight line is shown in Fig. 6(d).

(ii) Low DC offset

In presence of low amount of DC offset less than the signal amplitude, $s(t)=\sin(2\pi t)+0.5$, the origin is still located in the Hilbert transform circle but is not located exactly on the centre of the circle (red circle in Fig. 6(b)). Thus, for a point turning on the circle with constant ω , the phase slope is time varying. It may be seen that from the blue curve in Fig. 6(c) that the slope is always positive, therefore the corresponding instantaneous frequency possesses some positive peaks in major phase variations.

(iii) High DC offset

In this case, $s(t) = \sin(2\pi t) + 2$, the origin is located out of the Hilbert circle (brown circle in Fig. 6(b)). Therefore, for a moving point with constant ω the instantaneous phase possesses negative slope in each period. These negative slopes in instantaneous phase diagram cause negative values in the corresponding IF (brown curve in Fig. 6(d)).”

Reviewer#3: I am happy with the changes made. I have one minor point - in the conclusions the point is made that the performance of the MOV method is better than the comparison below 18dB SNR. It would be useful to indicate if this is within the range typically seen for ECG recordings.

Author reply: It is difficult to measure SNR in ECG signals precisely as one is not perfectly capable of distinguishing signal from noise after recording. In addition, the MoV approach outperforms the approach of Benitez et al in all conditions; moreover, it shows superior performance when we intentionally add noise to the signals.

Author action: We have now altered the last sentence in the Conclusion “For 18 dB SNR or less, the MoV approach shows reduced error.” To now read “The MoV approach demonstrates reduced error especially for 18 dB SNR or less. The result also suggests that MoV may be considered as an alternative method to IF or other instantaneous parameters in other non-stationary applications, particularly in noisy conditions.”

Appendix C

Manuscript ID: RSOS-182001.R1

Title: On moment of velocity (MoV) for signal analysis

Author: Dorraki *et al*

To: Prof. R. Kerry Rowe, Editor

Dear Kerry,

Many thanks for the valuable comments on our paper RSOS-182001.R1. We have pleasure in attaching an updated manuscript. The paper has been thoroughly proof read by a native English speaker, for grammar and clarity, and about 70 minor corrections have been carried out.

Best regards,

Mohsen Dorraki *et al*.